



# Climatological comparison of polar mesosphere summer echoes over the Arctic and Antarctica at 69°

Ralph Latteck[1] and Damian J. Murphy[2]

[1]Leibniz-Institut für Atmosphärenphysik, Kühlungsborn
[2]Australian Antarctic Division, Kingston,Tasmania, Australia

**Correspondence:** R. Latteck (latteck@iap-kborn.de)

**Abstract.** Polar Mesosphere Summer Echoes (PMSE) have been observed for more than 30 years with 50-MHz VHF radars at various locations in the Northern Hemisphere. Continuous observations of PMSE are conducted on the northern Norwegian island of Andøya (69.3°N) using the ALWIN radar (1999–2008) and MAARSY (since 2010). The same kind of PMSE measurements began in 2004 in the southern hemisphere with the Australian Antarctic Division's VHF radar at Davis Station in Antarctica (68.6°S), which is at an opposite latitude to Andøya. Since the radars at both sites are calibrated, the received echo strength of PMSE from more than one decade of mesospheric observations on both hemispheres could be converted to absolute signal power, allowing a direct comparison of the measurements. Comparison of PMSE observations obtained at both radar sites during a period of 23 boreal summers (Andøya) and 15 Austral summers (Davis) shows that their PMSE signal strengths are of the same order of magnitude but significantly less PMSE is observed in the southern hemisphere than in the northern hemisphere. Compared to Andøya, the PMSE season over Davis starts about 7 days later on average and ends 8 days earlier, making it 15 days shorter. PMSE over Davis occur less frequently but with greater variability in seasonal, diurnal, and altitudinal occurrence. For example, PMSE over Davis reach maximum altitudes about 1.5 km higher than those over Andøya.

## 1 Introduction

VHF radar echoes from the mesosphere and lower thermosphere have been detected from the Norwegian island of Andøya for more than twenty five years. The strong echoes from the mesopause region and above observed mainly in the summer months, have been known as Polar Mesosphere Summer Echoes (PMSE) for more than 30 years. PMSE result from inhomogeneities in electron density of a size comparable to the radar Bragg scale caused by neutral air turbulence combined with the action of negatively charged aerosol or ice particles, the latter existing only in the extremely cold mesopause region during the summer months. The existence of ice particles in the mesopause region or the resulting visual phenomenon of noctilucent clouds (NLCs) is known from various ground-, rocket-, and satellite-based observations. The close connection between PMSE and NLC was confirmed early on by simultaneous and common-volume lidar and radar measurements (e.g. von Zahn and Bremer, 1999). The early observations of PMSE and its relation to NLC are discussed in detail in Cho and Röttger (1997), while a comprehensive overview of the current understanding of this phenomenon is given in Rapp and Lübken (2004).





**Table 1.** Basic radar parameters and experiment configurations as used with ALWIN, MAARSY and the Davis VHF radar and relevant for the determination of volume reflectivity from PMSE observations as used in this study.

| radar | ALWIN | MAARSY | Davis VHF radar | | | |
|---|---|---|---|---|---|---|
| period | 1998–2008 | since 2011 | 2003/2004 | 2005/2006 | 2007 – 2012 | since 2014 |
| peak power $P_t$ | 36 kW | 736 kW | 20 kW | 36 kW | 60 kW | 38 kW |
| number of transmitting antennas | 144 | 433 | 144 | 144 | 144 | 144 |
| transmitting antenna gain $G_t$ | 28.3 dBi | 33.5 dBi | 28.9 dBi | 28.9 dBi | 28.9 dBi | 28.9 dBi |
| number of receiving antennas | 144 | 433 | 144 | 144 | 144 | 144 |
| receiving antenna gain $G_r$ | 28.3 dBi | 33.5 dBi | 28.9 dBi | 28.9 dBi | 28.9 dBi | 28.9 dBi |
| effective beam width (HPHW) $\theta_{[1/2]}$ | 2.12° | 1.27° | 2.12° | 2.12° | 2.12° | 2.12° |
| effective pulse width $\tau$ | 2 μs | 1.4 μs | 4 μs | 3 μs | 3 μs | 3.3 μs |
| system losses $e$ | 0.58 | 0.54 | 0.5 | 0.5 | 0.5 | 0.5 |
| → *system factor* $c_{sys}$ | $7.2 \cdot 10^{-9}$ | $1.3 \cdot 10^{-10}$ | $6.2 \cdot 10^{-9}$ | $3.9 \cdot 10^{-9}$ | $2.9 \cdot 10^{-9}$ | $3.83 \cdot 10^{-9}$ |
| number of coherent integrations | 32 | 32 | 116 | 104 | 52 | 118 |
| number of code elements | 16 | 16 | 1 | 8 | 8 | 1 |
| receiver gain | 101 dB | 101 dB | 80 dB | 80 dB | 80 dB | 70 dB |
| → *calibration factor* $c_s$ | $5.8 \cdot 10^{-20}$ | $1.6 \cdot 10^{-17}$ | $2.2 \cdot 10^{-22}$ | $2.1 \cdot 10^{-20}$ | $5.6 \cdot 10^{-20}$ | $5.7 \cdot 10^{-21}$ |

However, suitable measurements in the Southern Hemisphere (SH) have been rare in the past and limited to low southern
latitudes, largely due to the lack of radars in the Southern polar region. The first experiments for PMSE observations in the
southern hemisphere were conducted by Balsley et al. (1993) at the Peruvian Antarctic base on King George Island (62.1°S)
called Machu Picchu, during the Austral summer of 1992/1993. Analysis of these observations led to the conclusion that PMSE
do not exist at this latitude (Balsley et al., 1993) or, if they exist at all, these echoes must be at least 34 to 44 dB weaker than
their NH counterparts (Balsley et al., 1995). A year later, the first SH PMSE were observed with the improved radar at Machu
Picchu (Woodman et al., 1999). These observations confirmed the earlier conclusions that there are large differences between
the strength of PMSE observed in the two hemispheres. Woodman et al. (1999) attributed this asymmetry to differences in
mesopause temperature between the two sites. Huaman and Balsley (1999) suggested that differences in water vapour and
dynamics may be the cause of the observed delay in the occurrence of PMSE. However, Lübken et al. (1999) showed, based
on in situ measurements, that there is no significant difference in polar mesopause temperature between the two hemispheres.
Lübken et al. (2017), on the other hand, showed with high-resolution temperature measurements using resonance lidar at Davis
that a sudden mesopause height increase and associated mesopause temperature decrease can occasionally occur in the SH.
These so-called "mesopause jumps" only occur in the SH and are associated with the late breakdown of the polar vortex.

Morris et al. (2004) presented the first morphology of daily and seasonal occurrence of SH-PMSE based on VHF radar
observations at Davis during 2003/2004 Austral summer. They concluded that SH-PMSE observed at 68.6°S had similar char-



acteristics to published observations (i.e., height, intensity, daily, and seasonal distribution of occurrence) at similar northern latitudes, at least for the last three weeks of the Austral PMSE season. Latteck et al. (2007) conducted the first comparison of continuous measurements of PMSE collected at Andøya (69°N) during the 2004 boreal summer and at Davis (69°S) during 2004/2005 Austral summer based on radar volume reflectivity. They found that PMSE observed at Davis were weaker and reached maximum altitudes about 1 km higher than those observed in the Northern Hemisphere at an equivalent latitude. In ad-

dition, PMSE over Davis occurred less frequently but with greater variability. The PMSE seasons studied began about 34 days before the solstice at Andøya and Davis, but the duration of the PMSE season was about 9 days shorter at Davis. Another study that examined continuous PMSE observations during three Arctic summers at Andøya and three Antarctic summers at Davis confirmed these statements (Latteck et al., 2008). The volume reflectivity distribution of PMSE observed at Andøya showed a larger maximum ($\sim 2 \cdot 10^{-9}\,\text{m}^{-1}$) than the distribution of its counterparts observed at Davis ($\sim 4 \cdot 10^{-11}\,\text{m}^{-1}$). The mean

PMSE occurrence was smaller and more variable over Davis than at Andøya. The duration of the mean PMSE season was about 16 days shorter at Davis than at Andøya. The diurnal variations in PMSE occurrence showed a maximum at 11–16 LMT in both hemispheres. The mean altitude was 85.5 km at Davis, about 0.7 km higher than at Andøya. The vertical extent of the PMSE height distribution was 8.4 km over Davis, about 3 km less than at Andøya. Differences in mesospheric temperatures were suggested as a major cause of the observed differences in PMSE occurrence at Davis and Andøya, as shown by model studies and

supported by temperature measurements by meteor radars. Kirkwood (2007) compared PMSE measured with cross-calibrated VHF radars in the Arctic (Kiruna, 68°N) and Antarctica (Wasa, 73°S) and found that the PMSE characteristics of the two sites were very similar in late summer.

In this paper, the interhemispheric PMSE comparison is revisited by including more than one decade of observations. It is confirmed that the SH PMSEs are indeed more climatologically variable in terms of season, time of day and altitude than their

NH counterparts. The paper is structured as follows. The observations are described in section 2, followed by the climatological comparisons of reflectivity and distribution of occurrence as a function of season, time of day and altitude in section 3, a discussion (section 4) and summary.

## 2  Observation of polar mesospheric summer echoes at Andøya (69.3°N) and Davis (68.6°S)

Mesospheric radar echoes have been observed at the northern tip of Andøya, Norway (69.3°N, 16.0°E), starting from the early

1990s using the mobile SOUSY radar (Czechowsky et al., 1984). Since 1994, these observations have been continued during summer measurement campaigns with the ALOMAR SOUSY radar (Singer et al., 1995). ALOMAR SOUSY was replaced in 1999 by the ALWIN radar (Latteck et al., 1999), which provided continuous operation between 1999 and 2008. ALWIN was followed in 2010 by the Middle Atmosphere Alomar Radar SYstem MAARSY (Latteck et al., 2012). During the construction phase of MAARSY in 2009, ALWIN was operated with a reduced number of antennas for PMSE observations (Latteck et al.,

2010). In the summer of 2010, the observation of mesospheric echoes could be continued with a first expansion stage of MAARSY (Latteck et al., 2010). Since May 2011 MAARSY is fully extended. Both VHF radar systems on Andoya were





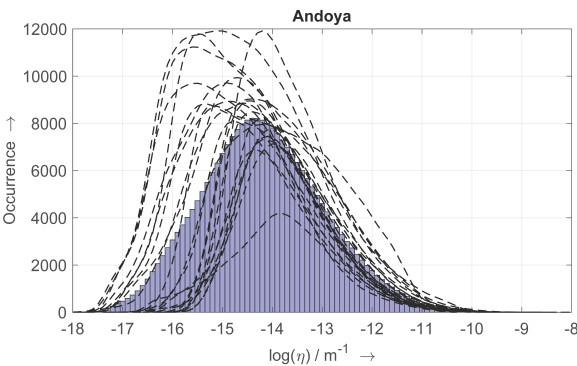
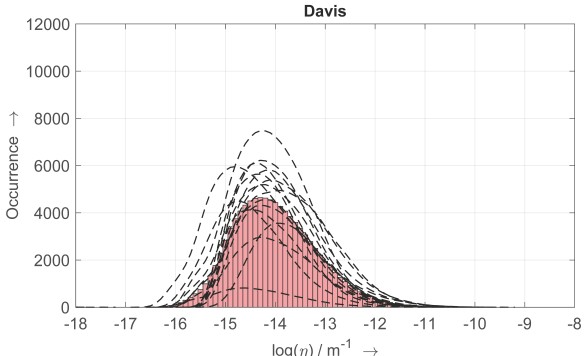

**Figure 1.** Annual distributions of volume reflectivity (dashed lines) of mesospheric echoes determined over Andøya (1999–2022, left panel) and Davis (2005–2022, right panel). The blue and red bars represent the mean values, respectively

.

operated on a radar frequency of 53.5 MHz. Most of the other technical parameters such as the transmitter peak power, the filter characteristics of the receivers, or the antenna aperture were different (Tab. 1).

The VHF radar at Davis, Antarctica (68.6°S, 78.0°E) was installed late in the austral summer of 2002/2003 (Morris et al.,
2004). Apart from the radar frequency of 55.0 MHz and the slightly lower peak power (see Tab. 1), the Davis MST radar was similar to ALWIN in its design and technical parameters at the time of its installation. In January 2005, the Davis VHF radar received a system upgrade, mainly replacing the transmitter and beam control unit. The latter was changed another time in 2014. Since that year, the radar was also not operated in any special mode for mesospheric observations. Instead, the focus was placed on tropospheric measurements. This was expressed among other things in an increased pulse repetition frequency (4750
Hz) adapted to range-alias the PMSE altitude range such that mesospheric echoes were recorded between approx. 17 to 30 km.

In order to use comparable parameters from data from different radars, the received echo power was converted to radar volume reflectivity. The radar volume reflectivity $\eta$ is defined as the power that would be scattered if all powers were isotropically scattered with a power density equal to that of the backscattered radiation, per unit volume and per unit incident power density (Hocking, 1985). It can be expressed as

$$\eta = \frac{P_r \, 128 \, \pi^2 \, 2 \ln(2) \, r^2}{P_t \, G_t \, G_r \lambda^2 \, e \, \theta_{[1/2]}^2 \, c \, \tau} \qquad (1)$$

where $r$ is the distance to the scatterers, $G_t$ and $G_r$ are the one-way gain of the transmit and receive antennas, respectively, $\theta_{[1/2]}$ is the effective half-width of the combined transmit/receive antenna beam, $\lambda$ is the radar wavelength, $e$ is the system efficiency, which mainly includes the antenna feed system losses, $P_t$ is the transmitted peak power, $P_r$ is the received signal power, $c$ is the speed of light, and $\tau$ is the effective pulse width (Hocking and Röttger, 1997). The correction term $2 \ln(2)$
accounts for the non-uniform antenna gain over half the power width (Probert-Jones, 1962; Skolnik, 1990). All the system-dependent parameters of Eq. 1 can be combined into a system factor $c_{sys}$, as shown in Table 1 for the different periods and





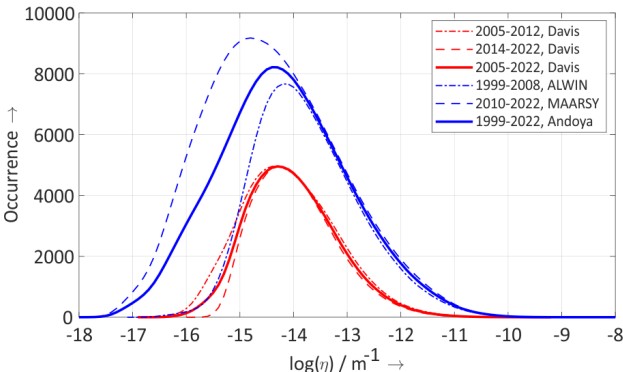

**Figure 2.** Mean distributions of PMSE volume reflectivity obtained by ALWIN and MAARSY at 69°N (blue) during the boreal summer periods (May–August) 1999–2008 and 2010–2022 respectively, as well as obtained with the Davis VHF radar at 69°S (red) during the austral summer periods (November–February) 2005–2012 and 2014–2022.

radar configurations. Thus, the radar reflectivity $\eta$ depends only on the distance to the scatterers $r$ and the absolute value of the received signal power $P_r$

$$\eta \;=\; P_r \cdot c_{sys} \cdot r^2 \tag{2}$$

The VHF radars at Andøya and Davis are calibrated regularly and specifically before and after all major engineering changes following a procedure described in (Latteck et al., 2008) and Appendix A1. A compilation of the main technical parameters of the radars used in this study and the parameters of the experimental configurations as used for standard observations of PMSE are listed in Tab. 1. However, some of the parameters listed there have changed in experiment configurations during special campaigns.

## 3   Climatology of PMSE observed at Andøya (1999–2022) and Davis (2005–2022)

### 3.1   Signal strength characteristics

The PMSE datasets reused in this study are based on 6-minute averages of radar volume reflectivities of PMSE observations from ALWIN and MAARSY and the Davis VHF radar, respectively. To use reliable values and exclude outliers, PMSE events were detected and flagged in the datasets. A PMSE event was defined as a radar reflectivity increase above the detection limit, persisting for a minimum duration of 24 minutes (equivalent to 4 consecutive averages) within a single range gate.

Fig. 1 shows the annual distributions of volume reflectivity of PMSE obtained by ALWIN (1999–2008) and MAARSY (2011–2022) at 69°N (left panel) during the boreal summer periods (May–August) and by the Davis VHF radar (2005–2022) at 69°S (right panel) during the austral summer periods (November–February). The differences in minimum signal detectability,





**Table 2.** Extreme values and quantiles of mean annual distribution of volume reflectivity of PMSE obtained by ALWIN (1999–2008), MAARSY (2010-2022) and the Davis VHF radar during the periods 2005–2012 and 2014 – 2022.

| $\mathrm{m}^{-1}$ | Davis 2005-2012 | Davis 2014-2022 | ALWIN 1999-2008 | MAARSY 2010-2022 |
|---|---|---|---|---|
| $\eta_{min}$ | $2.5 \cdot 10^{-17}$ | $2.0 \cdot 10^{-16}$ | $1.6 \cdot 10^{-17}$ | $1.3 \cdot 10^{-18}$ |
| $\eta_{peak}$ | $5.0 \cdot 10^{-15}$ | $6.3 \cdot 10^{-15}$ | $7.9 \cdot 10^{-15}$ | $1.6 \cdot 10^{-15}$ |
| $\eta_{max}$ | $3.2 \cdot 10^{-10}$ | $2.0 \cdot 10^{-10}$ | $6.3 \cdot 10^{-09}$ | $4.0 \cdot 10^{-09}$ |
| $Q_{0.01}$ | $1.6 \cdot 10^{-16}$ | $6.3 \cdot 10^{-16}$ | $2.5 \cdot 10^{-16}$ | $1.3 \cdot 10^{-17}$ |
| $Q_{0.50}$ | $7.9 \cdot 10^{-15}$ | $1.0 \cdot 10^{-14}$ | $1.3 \cdot 10^{-14}$ | $3.2 \cdot 10^{-15}$ |
| $Q_{0.99}$ | $1.6 \cdot 10^{-12}$ | $1.3 \cdot 10^{-12}$ | $6.3 \cdot 10^{-12}$ | $4.0 \cdot 10^{-12}$ |

which determines the left-hand slope of the distributions, are in the case of ALWIN and MAARSY mainly determined by the

size of the antenna array used and the differences in peak power but are in general also affected by fluctuations caused by changes in the radar experiment configurations, especially by the use of different receiving antenna configurations and the number of coherent integrations as often used within the observation periods for specific campaigns (Latteck and Bremer, 2017). The annual distributions of Davis volume reflectivity (right panel) show only a small variation. This is mainly due to the fact that other than changes to the beam steering method to hardware-only, the antenna was not changed during the entire

period of the observations studied. Changes in other system parameters were accounted for by the system factor in Eq. 1 and receiver calibration.

The red and blue bars in Fig. 1 represent the mean distributions of volume reflectivity of PMSE obtained at Davis (right panel) and Andøya (left panel), respectively. In Fig. 2, these mean annual distributions of the volume reflectivity of PMSE (solid lines) determined at both sites are compared in one diagram. The additional curves are shown for Andøya and Davis for

the periods of major technical changes to the radar systems at both sites. Because the antenna size of ALWIN and the Davis VHF radar were identical and the other system parameters had comparable magnitudes (Tab. 1), the two systems had almost directly comparable minimum detection sensitivities (dashed-dotted lines in Fig. 2).

The mean distribution of the volume reflectivity of the observed PMSE covers a range beginning at the detection limit of the radars and extending to a maximum value of about $10^{-10}\mathrm{m}^{-1}$ for the Davis radar and $10^{-09}\mathrm{m}^{-1}$ for ALWIN and MAARSY,

with peak values of about $6.0 \cdot 10^{-15}\mathrm{m}^{-1}$ and $7.9 \cdot 10^{-15}\mathrm{m}^{-1}$ for Davis and ALWIN, respectively. The peak value of the distribution for MAARSY is lower at $1.6 \cdot 10^{-15}\mathrm{m}^{-1}$ because of the greater sensitivity of the system. The differences that can nevertheless be seen in the comparison of the left-hand slopes of the results from the Davis radar (red curves in Fig. 2) are due to an increased background signal in the tropospheric region in the determination of PMSE events compared to the direct measurement in the mesosphere, which in turn is due to the aliased PMSE measurement within a tropospheric experiment at

Davis from 2014.

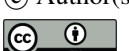



**Figure 3.** Seasonal height distribution and variation of PMSE occurrence and their diurnal occurrence variation above Andoøya (1999–2022, left panels) and Davis (2005–2022, left panels). The occurrence rates refer to radar reflectivities above a common threshold of $10^{-15}\mathrm{m}^{-1}$, and to the number of 6-minute averages per day (maximum 240). The seasonal height distribution of PMSE in the top plots are normalized to its maximum. The seasonal occurrence rates in the middle plots are based on the occurrence of a PMSE event at one height within a 6-minute time bin. The solid lines represent a 3-day running mean value. The bottom plots show the daily occurrence rate of PMSE normalized to its maximum.

The volume reflectivity quantiles, which are summarized in Tab. 2 quantitatively confirm the above; 1% of the echoes detected by the Davis radar in 2005–2012 period ($Q_{0.01}$) have a signal strength of $\eta \leq 1.6 \cdot 10^{-16}\mathrm{m}^{-1}$, in the period 2014–2022 it is $\eta \leq 6.3 \cdot 10^{-16}\mathrm{m}^{-1}$. Similarly, the respective median values $Q_{0.5} = 7.9 \cdot 10^{-15}\mathrm{m}^{-1}$ change after 2012 to $Q_{0.5} = 1.0 \cdot 10^{-14}\mathrm{m}^{-1}$. The value $Q_{0.01}$ of the PMSE observations from ALWIN is $\eta \leq 2.5 \cdot 10^{-16}\mathrm{m}^{-1}$, which is close to that of the 2005–2012 observations from Davis. The same is true for $Q_{0.5} = 1.3 \cdot 10^{-15}\mathrm{m}^{-1}$ as the median value of the ALWIN-PMSE distribution, illustrating once again the direct comparability of the two systems. MAARSY's values are correspondingly lower with $Q_{0.01} = 1.3 \cdot 10^{-17}\mathrm{m}^{-1}$ and $Q_{0.5} = 3.2 \cdot 10^{-15}\mathrm{m}^{-1}$ due to the higher sensitivity of the radar. Interesting are the differences in

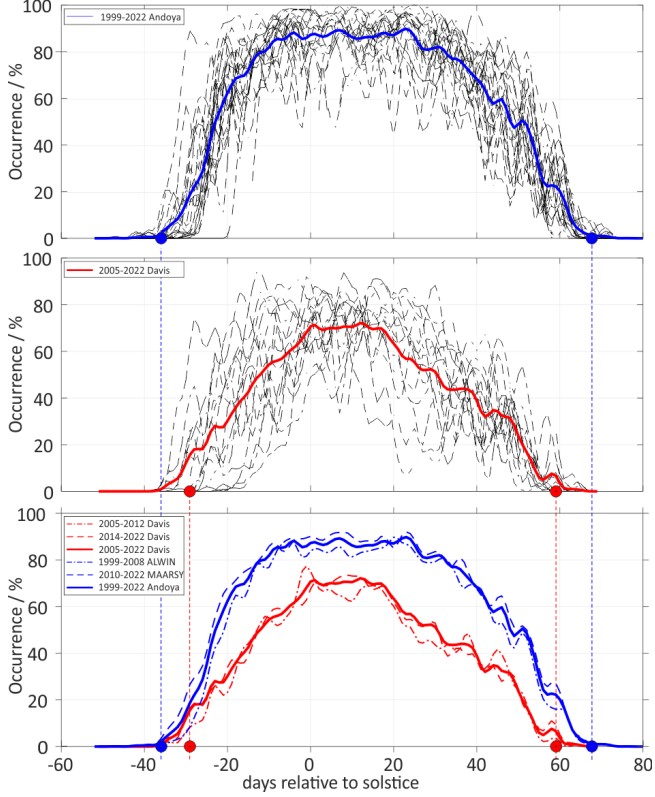

**Figure 4.** Seasonal variations in the frequency of occurrence of PMSE in Andøya, $69°$N (top panels) and Davis, $69°$S (middle panels). The blue and red solid curves represent the mean values of the occurrence rates over the entire observation period for Andøya and Davis, respectively. The bottom panel compares the mean occurrence rates for the PMSE observations in Andøya and Davis for different time periods. The blue and red dots mark the mean start day and the mean end day of the respective PMSE seasons.

the maximum values $\eta_{max}$ and $Q_{0.99}$ of the PMSE distributions of the northern and southern hemispheres. If the differences in $\eta_{max}$ are in the range of one order of magnitude, they are still different by a factor of about 4 in the statistical $Q_{0.99}$.

## 3.2   Seasonal, diurnal and altitudinal occurrence distributions

The minimum signal detectability of the radars used in this study is caused not only by different system characteristics as listed in Tab. 1 but also by changes in radar experiment configurations, mainly by a change in the number of coherent integrations used within observing seasons for specific campaigns. To obtain as unbiased a picture as possible of the seasonal and daily variations in the occurrence rates of polar mesospheric summer echoes, a lower threshold of $\eta_{min} = 10^{-15}$m$^{-1}$ was set. This threshold is below the peak $\eta_{peak}$ of the PMSE distribution from ALWIN and the Davis VHF radar as listed in Tab. 2, but is also larger than the minimum volume reflectivity detected by all radars at Davis and Andøya, and thus should cause nearly all detected PMSE to have been counted regardless of variations in the minimum detection limit. The chosen threshold of $\eta \geq 10^{-15}$m$^{-1}$



**Table 3.** Earliest, mean, and latest onset and end of PMSE season in Andøya (1999–2022) and Davis (2005–2022) for signal strengths $\eta_{min} = 10^{-15} \text{m}^{-1}$.

| PMSE season | | Davis | | Andøya | |
|---|---|---|---|---|---|
| | | day/rts. | date | day/rts. | date |
| begin | earliest | -37 | Nov 14 | -47 | May 05 |
| | mean | -29 | Nov 21 | -36 | May 16 |
| | latest | -21 | Nov 30 | -18 | Jun 03 |
| end | earliest | 50 | Feb 09 | 57 | Aug 17 |
| | mean | 59 | Feb 18 | 67 | Aug 27 |
| | latest | 65 | Feb 24 | 76 | Sep 05 |
| mean | duration | 89 d | | 105 d | |

also allows for qualitative comparison to other studies (e.g. Kirkwood et al., 2007; Latteck and Bremer, 2013, 2017; Sato et al., 2017; Latteck et al., 2021).

### 3.2.1 Seasonal variation of PMSE occurrence

Fig. 3 shows the mean seasonal and diurnal variation in the occurrence frequency of PMSE measured over Andøya (1999–2022) and Davis (2005–2022). The top panels show the mean frequencies of all echo detections normalized to the seasonal maximum value over time and altitude with respect to the daily measurement period. This plot allows a comparison of the seasonal dynamics of PMSE occurrence over the entire altitudinal range of observations. A qualitative comparison of mean absolute daily occurrence frequencies is provided in the middle panels of Fig. 3. Here, an occurrence is triggered if it occurs at any altitude. June/July and December/January are the months with the highest PMSE frequencies in the northern and southern hemispheres, respectively. During these periods, PMSE were observed with an average frequency of 82.2% over Andøya but only 55.3% over Davis. The shape of the mean PMSE frequency distribution in the left middle plot in Fig. 3 is similar and directly comparable to the results of previous studies of PMSE occurrence over Andøya (Bremer et al., 2009; Latteck and Bremer, 2013, 2017; Latteck et al., 2021).

The PMSE season, analyzed here based on signal strengths $\eta \geq 10^{-15} \text{m}^{-1}$, begins in the northern hemisphere on mean day -36 (May 16) and lasts to mean day 67 (August 27) relative to the summer solstice (rts) with a standard deviation of 6.0 and 4.4 days, respectively. One six-minute single-range occurrence is sufficient to trigger the presence of PMSE in this context. The earliest start of the PMSE season over Andøya during the observation period was recorded on day -47 rts (May 5), the latest start on day -18 rts (June 3). The earliest end of the season was on day 57 rts (August 17), the latest end on day 76 rts (September 05). The earliest start of the PMSE season over Davis during the observation period was on day -37 rts (November 14), 10 days later than over Andøya, and the latest start was on day -21 rts (November 30) which is only 3 days later compared to Andøya. The

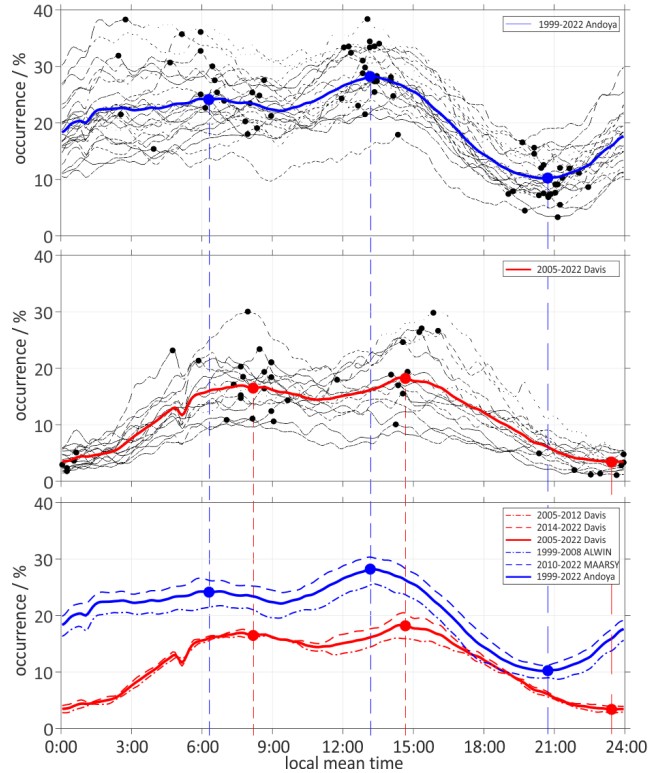

**Figure 5.** Mean diurnal variations of PMSE occurrence frequencies obtained at Andøya, 69°N (81–89 km, top panels) and Davis, 69°S (83–91 km, middle panels). The blue and red solid curves represent the mean values of the occurrence rates over the entire observation period. The bottom panel compares the mean occurrence rates for the PMSE observations at Andøya (blue curves) and Davis (red curves) for different time periods. The blue and red dots mark the daily times of the first and second maxima as well as the minima of the mean daily occurrence of PMSE over Andøya and Davis, respectively. The black dots indicate the corresponding times of the individual seasons.

earliest end of the season in the Southern Hemisphere was on day 50 rts (February 9) 7 days earlier than over Andøya, the latest end on day 65 rts (February 24), 11 days earlier than over Andøya), which is shown by the vertical thin dashed blue and red lines in Fig. 4. Thus, the mean start of the PMSE season in the southern hemisphere is day -29 rts (November 21) and the mean end is day 59 rts (February 18) relative to the solstice, with standard deviations of 5.6 and 4.0 days, respectively. This results in a significantly shorter mean PMSE season of 89 days in the Southern Hemisphere compared to 105 days in the Northern Hemisphere, as illustrated by the curves in Fig. 4 and summarized in Tab. 3.

### 3.2.2 Diurnal variation of PMSE occurrence

The mean daily variation in PMSE occurrence, shown in the bottom panels of Fig. 3, have similar characteristics in both data sets. PMSE at both sites occur clustered in the morning and midday hours in a range between 82 and 90 km, with the diurnal pattern of this occurrence showing differences in the comparison of the two observations. For illustration and quantification,





Fig. 5 compares the multi-season mean diurnal variation of both data sets. For this purpose, the mean value over the altitude range 81–89 km was used for the measurements from Andøya and the range 83–91 km was used for the measurements from Davis. The mean daily PMSE occurrences over Andøya shown in the upper part of Fig. 5 illustrate a pronounced pattern with large variation around a mean of about 20 % between midnight and 17:00 LMT. All mean trajectories also show a pronounced maximum between 11:00 and 15:00 LMT, peaking on average around 13:09 LMT. In the period between 17:00 and midnight, all mean diurnal traces show a pronounced minimum. In the 23-year mean response (blue curve), the frequency rate drops sharply during this period to about 10 % at 20:42 LMT and then rises again to 20 % by about 01:30 LMT. The morning course of the mean diurnal cycle shows a slight first maximum at 06:17 LMT, which is very weak, indicated by the strong variation of the maxima of the individual annual cycles (black dots) around this time.

Such a first maximum is far more prominent in the individual as well as the mean annual diurnal cycle of PMSE frequency over Davis (middle panel in Fig. 5). The mean diurnal cycle (red curve) begins with a minimum of about 3 % at 23:26 LMT near midnight then rises to the first maximum of about 16 % at 08:10 LMT, then drops somewhat to rise to about 18 % at 14:40 LMT for the second time, and then drops continuously until midnight.

Direct comparison of the mean diurnal variation of PMSE occurrences from Andøya and Davis in the bottom panel of Fig. 5 shows a clearly asymmetric pattern in the Andøya data compared to Davis. Also clearly visible is a larger offset of the second maximum at local noon of 2.65 h in Davis compared to 1.16 h in the observations from Andøya. The distance of the local minima to each other is even larger by about 2.7 h.

### 3.2.3 Altitudinal distribution of PMSE occurrence

Fig. 6 shows the altitude distributions of the PMSE occurrence rates of the individual measurement periods (dashed black lines) and their mean values (solid lines) for Andøya (left panel) and Davis (middle panel). The right panel directly compares the mean values for different measurement periods for both sites. The thickness of the distributions at the half maximum (stars in the right panel of Fig. 6) is ∼ 6 km at both sites. The peak of the PMSE altitude distributions of the individual years are marked with black dots in Fig. 6. The mean peak of the PMSE altitude distributions over Davis is at an altitude of 86.1 km (red dots) about 1.5 km higher than at Andøya site (84.6 km, blue dots).

The individual profiles of the PMSE altitude distributions for Andøa are characterized by a nearly Gaussian shape, while for Davis they are not. The latter indicates an uneven height distribution of PMSE frequencies, which is also evident in the upper right panel of Fig. 3, and which is subject to year-to-year variation. This fact is even more evident in the individual and climatological seasonal variations of the peak altitudes of PMSE occurrences, as shown in Fig. 7. The individual seasonal variations of peak altitudes over Andøya (dashed lines in the top panel of Fig. 7) show much less variation compared to peak altitudes over Davis (dashed lines in the middle panel of Fig. 7). The mean seasonal variations of peak heights of PMSE occurrence calculated for the mean duration of each PMSE season (blue and red curves), on the one hand, illustrate the difference in the mean distribution of peak heights already mentioned and, on the other hand, show a non-uniform peak separation in the annual variation. Both mean curves of the PMSE peak heights (lower panel in Fig. 7) show an almost even spacing of about 1.2 km over a period of about 20 days at the beginning of the PMSE season. About 10 days before the solstice,





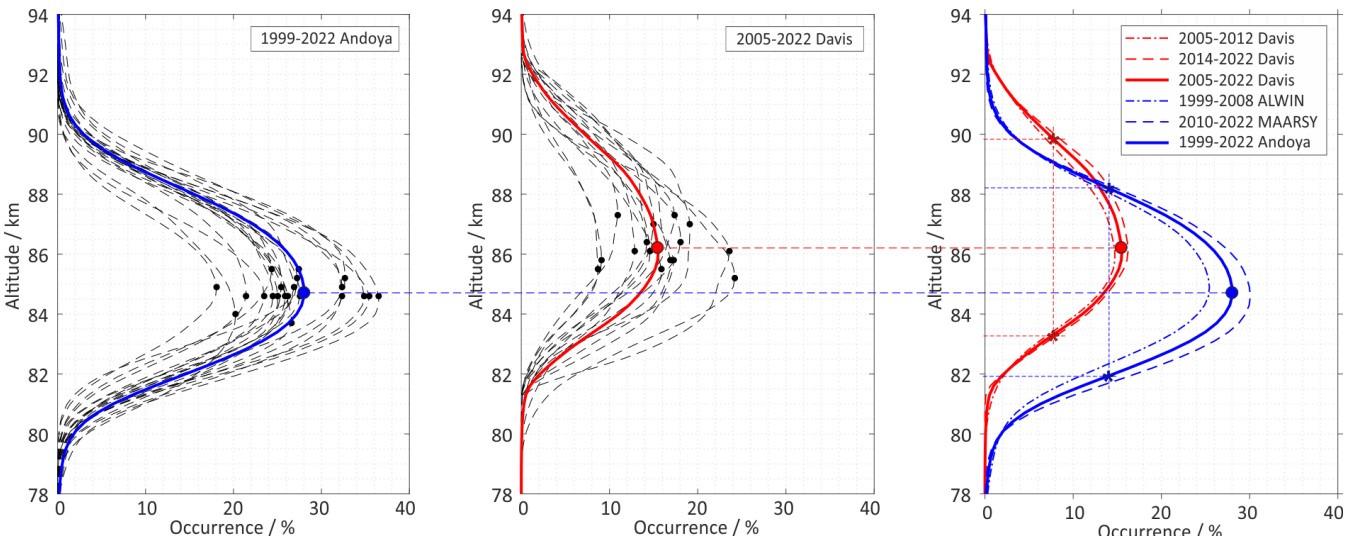

**Figure 6.** Mean annual altitude distributions of PMSE occurrences obtained at Andøya, 69°N (left panel) and Davis, 69°S (middle panel) during the boreal summer period (May–August) 1999-2022 and the austral summer period (November-February) 2005-2022, respectively. The blue and red solid curves represent the mean values of the altitude distributions of PMSE occurrences over the entire observation period. The right panel compares the mean height distributions of PMSE for the Andøya (blue curves) and Davis (red curves) observations for different time periods. The blue and red dots mark the mean peak heights of each PMSE occurrence, and the stars indicate the half-maxima of the distributions.

the mean PMSE peak altitude over Davis jumps upward by about 1 km, while over Andøya drops just before the solstice. As the season progresses, the PMSE peak altitude over Davis decreases uniformly while over Andøya remains nearly constant. About 20 days after solstice, the latter starts to increase slightly, so that the PMSE peak altitude at both sites has almost converged by 215  the end of the season.

## 4   Discussion

This paper gives an overview of continuous measurements of polar mesospheric summer echoes obtained with the VHF radars ALWIN (1999-2008) and MAARSY (2011-2022) on the northern Norwegian island of Andøya (69.30°N, 16. 03°E) during 23 years of boreal summers and with the VHF radar at Davis (68.6°S, 78.0°E) during 15 years of austral summers (2005–2012 220  and 2014–2022). The fact that the radar systems at both sites have been regularly calibrated results in a unique data set that allows PMSE to be analysed and compared in terms of seasonal, diurnal and altitudinal occurrence distributions, as well as in signal strength of the received backscattered echo.





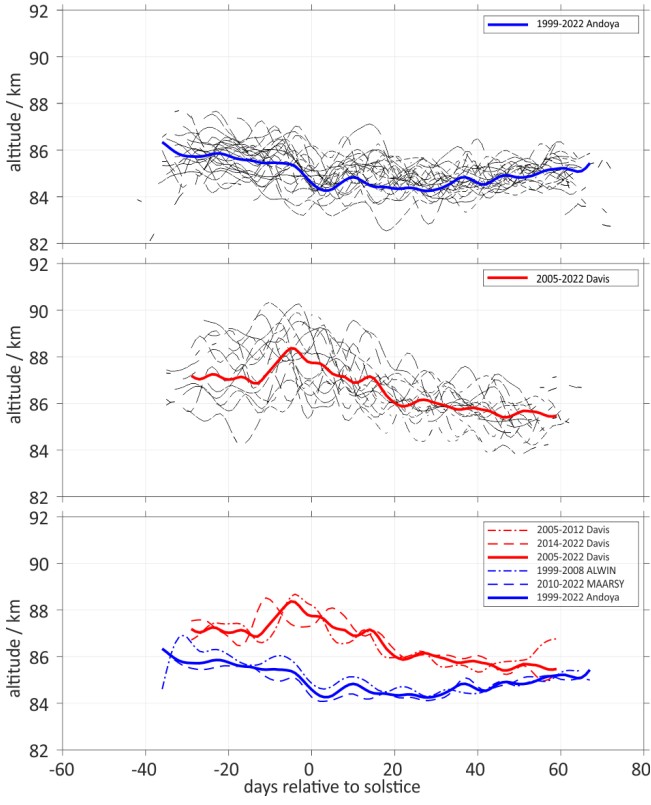

**Figure 7.** Seasonal variations in mean altitude of PMSE occurrence at Andøya, 69°N (top panel) and Davis, 69°S (middle panel) during the boreal summer period (May–August) 1999-2022 and the austral summer period (November-February) 2005-2022, respectively. The blue (Andøya) and red (Davis) solid curves represent the mean seasonal variation in peak height of PMSE occurrence over the entire observation period. The bottom panel compares these mean variations for the observations in Andøya and Davis for different time periods.

## 4.1 Signal strength of PMSE

Since ALWIN and the Davis VHF radar are comparable both in the size of the antenna and in many technical parameters,
the distributions of the signal strength (Fig. 1) from the calibrated received signals of the mesospheric echoes are directly comparable in terms of their maximum values. Due to the system-related differences between the instruments (Tab. 1), the changes in the experiment setup in the course of the respective measurement campaigns, and the enhanced performance of MAARSY, the minimum signal detectability is not constant over the entire measurement period considered. This leads to differences in the increasing slopes of the individual as well as the mean distributions of the volume reflectivities (Fig. 2)
calculated from the received signals of the radars.

The mean distributions of the volume reflectivity of the observed PMSE cover a range starting at the detection limit of the radars and extending to a maximum value of about $1.45 \cdot 10^{-12} \mathrm{m}^{-1}$ for the Davis radar and $5.15 \cdot 10^{-12} \mathrm{m}^{-1}$ for ALWIN and MAARSY, in terms of all values larger than 99% (see $Q_{0.99}$ in Tab. 2). The ratio of the statistical maximum values of





the PMSE observations of the northern hemisphere to the southern hemisphere is thus less than a factor of 4. Differences in
observed maximum values of volume reflectivities in this order of magnitude were already found in the very first comparative
study of PMSE observations at Davis and Andøya (Latteck et al., 2007), despite being based on only one measurement season
of the Davis radar before its modification at the end of January 2004. The very strong differences of about two orders of
magnitude between the Davis and Andøya PMSE observations described in Latteck et al. (2008), which are mainly based
on the first two measurement periods of the Davis radar after its reconstruction, are not confirmed here and are likely due to
inaccurate information about a system parameter in the receive path of the Davis radar after the modification at the time of the
study (see Appendix A1). The results of the present study also do not confirm the conclusion of the very first observations of
PMSE in the southern hemisphere (Woodman et al., 1999) that there are large differences between the strength of PMSE in the
two hemispheres, but confirm the earlier observation of Morris et al. (2004, 2006), Kirkwood (2007) and Latteck et al. (2007)
that the signal strength of PMSE in Antarctica is similar to that of PMSE in the Arctic.

**4.2 Seasonal, diurnal, and altitudinal variations of PMSE**

In order to obtain as unbiased a picture as possible of the seasonal and daily variations in the occurrence rates of the PMSE,
which could be caused by the system and experiment-related differences in the detection limit of the radars (Tab. 2), a minimum
threshold of $\eta_{min} = 10^{-15}\mathrm{m}^{-1}$ was set for investigating the seasonal and diurnal variation in the occurrence of PMSE in both
hemispheres. This threshold also allows a qualitative comparison of the current results with other studies (e.g. Kirkwood et al.,
2007; Latteck and Bremer, 2013, 2017; Sato et al., 2017; Latteck et al., 2021).

**4.2.1 Seasonal variation of PMSE occurrence**

The PMSE season in Davis (69°S) is more variable compared to Andøoya (69°N), both in terms of the length of the season
and the occurrence frequency within the season. This is mainly due to the longer extent and greater variability of the collapse
of the polar vortex in the SH compared to the NH (Lübken et al., 2015).
The shape of the mean seasonal variation of PMSE occurrence over Andøya (Fig. 3 middle left panel) is similar and directly
comparable to the results of previous studies (Bremer et al., 2009; Latteck and Bremer, 2013, 2017; Latteck et al., 2021).
The direct comparison with the Davis measurements in the lower part of Fig. 4 shows that significantly fewer PMSE were
observed in the southern hemisphere during the 15 years under consideration. A qualitative comparison of the mean absolute
daily occurrence frequency (Fig. 3, middle panel) shows that during the months with the highest PMSE occurrence frequencies
(June/July in the northern hemisphere and December/January in the southern hemisphere), PMSE was observed with an average
frequency of 82.2% over Andøya, but only 55.3% over Davis.
The PMSE season in the Southern Hemisphere starts on average 7 days later and ends 8 days earlier than in the Northern
Hemisphere, resulting in a significantly shorter mean PMSE season of 89 days in the Southern Hemisphere compared to 105
days in the Northern Hemisphere. This feature was already noted in an earlier study by Latteck et al. (2008) and attributed
to differences in the dynamic and thermal state of the mesopause region in both hemispheres. PMSE are present in both
hemispheres in summer as long as the equatorward winds transport cold air from higher to lower latitudes. These meridional





winds support the transport of PMSE particles from higher latitudes, as shown by 3-D modelling of the formation of noctilucent cloud particles (Berger and von Zahn, 2007). In this context, the shorter PMSE season of the SH is also reflected in the earlier transition of the meridional winds to winter conditions (Dowdy et al., 2001; Morris et al., 2006). The observed end of the

PMSE season over Davis at day 59 rts coincides with an increase in mesospheric temperature measured by Lübken et al. (2004) with falling spheres over Rothera (67°S) in January/February 1998. The observed shorter PMSE season and especially the earlier end in the southern hemisphere was also predicted by Lübken and Berger (2007) with the LIMA/ice model. The model reproduces the main PMSE characteristics observed by several VHF radars in both the northern hemisphere and southern hemisphere.

### 4.2.2 Diurnal variation of PMSE occurrence

The mean daily variation of PMSE occurrence shown in the lower plots of Fig. 3 show similar signatures but also pronounced differences in both data sets. PMSE occur more frequently in the morning and midday hours at both sites in a range between 82 and 90 km, although the diurnal pattern of this occurrence shows differences when comparing the two observations.

The mean daily occurrence of PMSE over Andøya (Fig. 5, blue curves) shows a distinct pattern with large fluctuations around

a mean value of about 20 % between midnight and 17:00 LMT. During this time, the pattern shows a pronounced maximum between 11:00 and 15:00 LMT, peaking on average at 13:09 LMT and then continuously decreasing towards a minimum of about 10 % reached at 20:42 LMT.

The pattern of mean daily PMSE occurrence over Davis (Fig. 5, red curves) is similar compared to Andøya observations, but the maxima and minima are shifted by 1.5 and 2.7 hours, respectively. The curve starts with a pronounced minimum of about

3 % at 23:26 LMT near midnight then rises to a clear first maximum of about 16 % at 08:10 LMT, drops a bit to rise a second time to about 18 % at about 14:40 LMT, and thereafter drops continuously until midnight.

Both the position of the absolute (second) maximum and the minimum of the Andøya observations are consistent with results of previous studies (e.g. Hoffmann et al., 1999; Bremer et al., 2001; Latteck et al., 2021). Hoffmann et al. (1999) compared the diurnal variations of PMSE signal strength (SNR) and the meridional wind component at PMSE heights and found that both

parameters have similar extreme values, but offset in time. On the assumption that meridional winds can transport cold air from high polar latitudes and vice versa, and that cold temperatures are one of the necessary conditions for the formation of PMSE, they concluded temperature changes induced by meridional tidal winds transporting cold (warm) air from polar (equatorial) latitudes to the observation site can have a significant impact on the diurnal variation of PMSE. Bremer et al. (2001) explained that the semi-diurnal variation of PMSE with maxima at noon and midnight was due to the influence of the diurnal variation of

geomagnetic activity.

### 4.2.3 Altitudinal variation of PMSE occurrence

The peak of the PMSE altitude distribution over Davis is 86.1 km, which is about 1.5 km higher than the corresponding value at Andøya (84.6 km) (Fig. 6). This is consistent with a higher and colder mesopause along with warmer temperatures at lower heights (Lübken et al., 2015). The absolute value is slightly more than the 1 km difference reported in Latteck et al. (2007),





which was in good agreement with observed polar mesospheric cloud (PMC) height differences (Wrotny and Russell III, 2006).
The analysis of HALOE data collected between 55° and 70° latitudes in both hemispheres revealed a mean height of the PMC
distribution in the Southern Hemisphere of 84.2 km, which is $0.9 \pm 0.1$ km higher than the mean height of the PMC distribution
in the Northern Hemisphere of 83.3 km. According to Wrotny and Russell III (2006), the observed interhemispheric differences
in the vertical extent of Polar Mesospheric Clouds (PMC) could be attributed to colder temperatures of 4–7 K, as seen in the
NH-HALOE temperature profiles between 75 and 86 km.

The thickness of the altitude distributions at the half maximum is about 6 km at both sites. The mean altitudes of the individ-
ual years show a significantly closer concentration around the mean value for the measurements above Andøya compared to the
corresponding values at Davis. This is also reflected in the annual distribution of the peak altitudes (Fig. 7). The mean PMSE
altitude observed in the NH shows an almost constant trend with a slight decrease towards the middle of the season shortly
after the solstice, followed by a slight increase towards the end of the season (Fig. 7, blue curves). In the SH, the mean trend
of peak heights tends to decrease over the course of the season but is significantly higher just before the solstice with ∼89 km
than during the periods before or after (Fig. 7, red curves). Lübken et al. (2017) attribute this increase to so-called "mesopause
jumps", an occasional sudden mesopause height increase and an associated mesopause temperature decrease, primarily, but
not only, observed prior and close to the solstice in the SH. The conditions for "mesopause jumps" are associated with the late
breakdown of the polar vortex when stratospheric winds are moderately eastward and mesospheric winds are strongly west-
ward. Under these conditions, gravity waves with comparatively large eastward phase speeds can pass the stratosphere and
propagate to the lower thermosphere because their vertical wavelengths in the mesosphere are rather large, implying enhanced
dynamical stability. When finally breaking in the lower thermosphere, these waves drive an enhanced residual circulation that
causes a cold and high-altitude mesopause (Lübken et al., 2017). The general downward trend of PMSE peak altitudes observed
over Davis specifically after the summer solstice is accompanied by the observed downward propagation of the mesopause in
the SH (Lübken et al., 2015), which was explained in Becker et al. (2015).

## 5   Summary and Conclusions

Continuous radar observations of the polar mesosphere were made at Andøya (69.30°N, 16.03°E) and Davis (68.6°S, 78.0°E)
during 23 years of boreal summers (May-August) 1999–2008 and 2011–2022 and 15 years of austral summers (November-
February) 2005–2012 and 2014–2022, respectively. This interhemispheric PMSE comparison incorporating more than a decade
of observations confirmed that the SH PMSEs are indeed more climatologically variable in terms of season, time of day and
altitude than their NH counterparts. The results of the studies of PMSE in terms of the variation of their seasonal and daily
occurrence as well as the signal strength of the received backscattered echo power can be summarized as follows:

–   PMSE signal strengths observed at locations of comparable latitudes in the northern and southern hemispheres are
330        similar. The maximum signal strengths of the measurements from Davis (69°S) have only a slightly lower peak volume
reflectivity of about $1.5 \cdot 10^{-12} \mathrm{m}^{-1}$ compared to $5.2 \cdot 10^{-12} \mathrm{m}^{-1}$ over Andøya (69°N).





– Fewer PMSE are observed in the southern hemisphere than in the northern hemisphere. The mean PMSE season based on 23 years of observations in Andøya, begins on May 16 (day -36 rts), lasts 105 days, and ends on August 27 (day 67 rts). In contrast, the mean PMSE season based on 15 years of observations at Davis begins on November 21 (day -29 rts), lasts 89 days, and ends on February 18 (day 59 rts). The average occurrence of PMSE for signal strengths $\eta \geq 10^{-15} \mathrm{m}^{-1}$ at 69°N in June/July is 82.3% while the average occurrence of PMSE in December/January at 69°S is 55.3%.

– The mean diurnal variation of PMSE frequencies in the Northern Hemisphere shows a nearly continuous course between midnight and about 17:00 LMT with a maximum at 13:09 LMT, which then leads to a sharp decrease to a pronounced minimum at 20:42 LMT. The mean diurnal pattern of PMSE frequencies in the Southern Hemisphere, on the other hand, shows a nearly symmetrical pattern, starting and ending with a minimum near midnight and two pronounced maxima at 08:10 LMT and 14:40 LMT.

– The altitude distribution of the PMSE occurrence in the southern hemisphere reaches its mean value at ∼86.1 km about 1.5 km higher than at Andøya.

– The mean PMSE altitude observed in the NH shows an almost constant trend with a slight decrease towards the middle of the season shortly after the solstice, followed by a slight increase towards the end of the season. In the SH the mean PMSE altitude tends to decrease over the course of the season but is significantly higher just before the solstice than during the periods before or after.

According to the results of the present study, the question raised by Balsley et al. (1993) "Southern-hemisphere PMSE: Where are they?" could be answered as follows: they can be found on average further south, and at higher altitudes, but are more variable than those over the northern hemisphere.

*Data availability.* The occurrence rates determined from the radar measurements, which were used to create the figures presented in this article, can be found in MAT data format at the following address https://www.radar-service.eu/radar/en/dataset/GoXVWUDVybjLlGyt?token=qEMvkFZAvIRuwfqhDMZu.

## Appendix A

### A1 Radar calibration

The VHF radar systems on Andøya (ALWIN and MAARSY) as well as the Davis VHF radar were regularly calibrated according to the methods described in Latteck et al. (2008). For this purpose, the signal of a calibrated noise source on the one hand and the attenuated transmit signal, on the other hand, were fed directly into the receiving system, which usually consists of a front-end amplifier, a baseband receiver and a digitizer. The basic setup of both methods is outlined in Fig. A1.



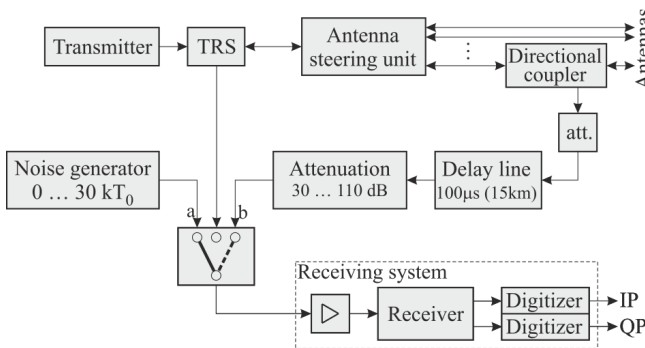

**Figure A1.** Principle of radar calibration (Latteck et al., 2008) using (a) a calibrated noise source, and (b) the delayed and attenuated transmitted signal.

Most of the calibration measurements of the Davis and the Andøya VHF radars were performed with the setup b shown in Fig. A1, since this leads directly to the calibration factor $c_s$ for coherently received signals. The transmit signal is taken from the antenna port using a directional coupler, delayed by $100\,\mu s$ using an ultrasonic delay line, corresponding to a virtual detection at 15 km at 53.5 MHz, and fed into the input port of the receiver. In the linear dynamic range of the receiver system, the calibration factor $c_s$ is obtained by comparing the output power $P_{s.out}$ in arbitrary units with the injected input power $P_{s.inp}$

in Watt

$$c_s = \frac{P_{s.inp}[W]}{P_{s.out}[au]} \tag{A1}$$

as listed in Table 1 for different measurements.

    When the Davis VHF radar was upgraded on 22 January 2005, changes were also made to the beam steering unit (BSU), which assigns the signals of the six transmitter outputs and six receiver inputs and associated phase offsets to the 36 antenna

groups for beam steering. These changes resulted in additional attenuation of the received signals, which is not captured by the calibration method described above. The resulting distribution of the determined volume reflectivities (black dashed line in Fig. A2) led to unrealistically small values compared to the measurements before the upgrade (black line in Fig. A2). This also becomes clear when comparing the solid black curve from Fig. A2 with the dash-dot curve in the left-hand figure of Fig. 1 in Latteck and Bremer (2017), which represents volume reflectivities of the ALWIN radar comparable at that time.

From the comparison of the mean values of the volume reflectivities greater than $Q_{0.99}$ of the distributions before the upgrade (black solid curve in Fig. A2) and the corresponding values for the same period of the 2005/2006 season (red dashed curve in Fig. A2), a correction factor to the calibration factor for after the upgrade was determined, which is the missing part describing the attenuation properties of the BSU. The solid red curve in Fig. A2 shows the distribution of PMSE volume reflectivities obtained from Davis PMSE measurements of one receive channel (24 antennas) with the corrected calibration

factor for 2004-2012.





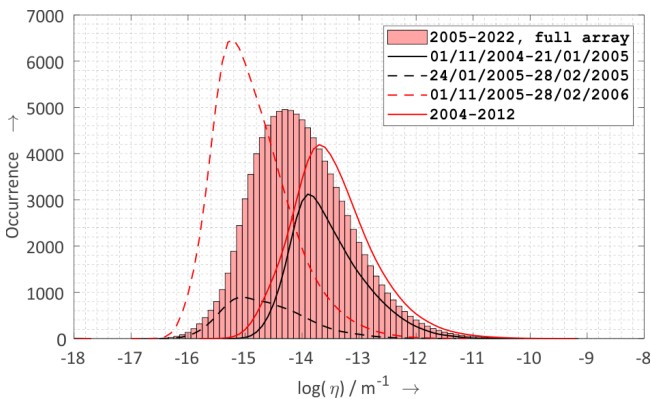

**Figure A2.** Mean distributions of PMSE volume reflectivity obtained with the Davis VHF radar at 69°S during the austral summer periods (November–February) of various periods using 24 out of 144 antennas for receiving (lines). The red bars represent the mean annual distribution (2005-2022) of PMSE volume reflectivity obtained with the Davis VHF radar using all 144 antennas for receiving as used in this study.

During the 2012/2013 PMSE season, further changes were made to the BSU on the Davis VHF radar, which resulted in the calibration method described above being applicable again. The comparison of the red curves in Fig. 2 shows the good agreement of the distributions after and before the 2012/2013 reconstruction. The red bars in Fig. A2 and the solid red curve in Fig. 2 represent the mean annual distribution (2005-2022) of PMSE volume reflectance obtained with the Davis VHF radar, but using all 144 antennas for the reception as used in this study.

*Author contributions.* RL and DJM had the main responsibility for the radar experiments in Andøya and Davis, respectively. RL analysed the data and wrote the article. DJM helped with the interpretation of the results and the discussion. Both authors read, corrected and agreed with the submitted version of the manuscript.

*Competing interests.* The authors declare no conflict of interest.

*Acknowledgements.* This work was supported by the Federal Ministry for Education and Research under grants 01 LG 1902A (project TIMA-2) in the frame of the Role of the Middle Atmosphere in Climate (ROMIC-II) program. The authors are indebted to the staff of Andøya Space for their permanent support. Radar operation at Davis was supported through Australian Antarctic Science projects 2325, 4025, 4445 and 4637. The authors thank David Holdsworth, Ray Morris and the wintering electronics engineers for their contributions to the Davis radar operation and optimisation.



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
