# Peer review of "Climatological comparison of polar mesosphere summer echoes over the Arctic and Antarctica at 69°"

_Annales Geophysicae, 2023_

## Author Response (AR1)

Dear Dr. Sato,

We would like to thank you for the effort you have put into the review process of our manuscript. We have uploaded into the system the responses to each reviewer and a manuscript with the tracking information for the changes. The revised manuscript incorporates changes recommended by the reviewers in their specific and technical comments.

In this paper, we have focused on the climatological analysis of radar observations, which has led to a statistically significant synthesis of previous studies based on much shorter data sets. We believe that the results presented represent a new level of knowledge due to the long data sets on which they are based, as they further minimise the uncertainties due to the short measurement times. Notably, we found that PMSE volume reflectivities at locations of comparable latitude in the NH and SH are similar, in contrast to previous findings.

We see our manuscript as a description of the long-term status of PMSE observations at both sites and at the same time as a basis for further studies that can address the differences that are now emerging. We agree with you, that the available data have the potential to attribute different physical origins to the observations. You allude to a relationship with solar cycle. Reviewer 3 enquires about the connection between PMSE and temperature, winds, water vapor etc. PMSE are a complex phenomena whose influences are multi-faceted. We believe that studies looking at the differences will need to access data sets that are not readily available (temperatures and water vapors beyond satellite passes, tidal fields beyond a single station). They will need to focus on analysing shorter time periods, which put them beyond the scope of this manuscript given the long observation time frame on which our research is based. Therefore, we have decided to limit the studies presented and the presentation of the results discussed to the scope indicated. We have responded to the third review with this in mind and hope that our arguments will also meet with your approval.

With kind regards,

Ralph Latteck and Damian Murphy

**Responses to the comments of Reviewer 1**

The authors would like to thank the reviewer for his/her positive assessment of the manuscript. In the revised manuscript, all notes listed under "Technical Comments" will be taken into account. We address the reviewer's specific comments below:

- *Upon thorough review, I noticed that there seems to be no mention of range resolution. Including this information somewhere in the paper would be beneficial.*

  The height resolution for all datasets studied is 300m. We will include this information in the text of the revised manuscript.

- *ll. 161-169: In addition to providing specific dates for the earliest and latest onset of the PMSE season, it might be beneficial to include the respective years as well. Doing so would offer readers a valuable reference when investigating inter-annual variations around the mesopause.*

  We thank the reviewer for this suggestion and will add the corresponding data to Table 3 in the revised manuscript.

- *ll. 287-295: Tidal effects are mentioned as one of the factors contributing to local time dependency of PMSE. On the other hand, previous studies, including Murphy et al. (2004), have pointed out seasonal variations in the amplitude and phase of tides in the Antarctic region. It would be valuable if the paper could discuss whether the local time dependency shown in lower panels of Figure 3 changes with the season, or at least remains qualitatively consistent. Such discussion could include how this relates to the research on the seasonality of tides.*

  This is indeed an investigation that we have not delved into in detail in the present study. Similar investigations/representations, as shown in the lower panel of Figure 3, which are based on monthly time periods, indicate, however, that the diurnal variation depicted in this figure does not exhibit significant seasonal (monthly) changes in the positioning of the maxima and minima, but rather annual changes in intensity. This could mean that any local time tidal influence on PMSE is due to migrating tides, which tend to be stable, rather than more variable non-migrating tides. Further investigation of this is thought to be beyond the scope of the paper, however, we will add this information without visual representation to Chapter 4.2.2 in the revised manuscript.

**Responses to the comments of Reviewer 2**

The authors would like to thank the reviewer for his/her positive evaluation of our submitted manuscript. We have limited the topic to the climatological study of radar observations, which now consolidates in a statistical sense the findings of earlier studies based on much shorter data sets. Among other things, however, it has also emerged that there are no differences in the volume reflectivity of the echoes observed in the southern hemisphere compared to the observations in the northern hemisphere, which has been a subject of speculation since the first publications in the early 1990s. Now that we have laid a climatological base with this study, we believe that further investigations could be carried out into deviations from the mean conditions, which could provide further insights into existing differences in the occurrence of these echoes, but this would go beyond the scope of the present study.

**Responses to the comments of Reviewer 3**

The authors thank the reviewer for the favourable assessment of our submitted manuscript, although some critical comments are made. We have focused on the climatological analysis of radar observations, which has led to a statistically significant synthesis that adds significantly to previous studies based on much shorter data sets. We believe that the results presented, due to the long data sets on which they are based, demonstrate a new level of knowledge, as they further minimise uncertainties due to short measurement times. They allow us to note that PMSE volume reflectivities at locations of comparable latitude in the NH and SH are similar, in contrast to previous findings. In our view, the fact that the possible causes or reasons listed are based on older studies or findings that are still valid does not detract from the knowledge gained.

We respect the reviewer's opinion and would like to point out that we see our manuscript as a description of the long-term status of PMSE observations at both sites and at the same time as a basis for further investigations that can address the differences that are now becoming apparent, e.g. in the seasonal and diurnal fluctuations in PMSE frequencies. In particular, the mentioned minimum in the diurnal variation of the NH observations, which is significantly weaker and time-shifted in the SH observations, could be of great interest.

We are of the opinion that the available data material has great potential to be able to assign various physical origins to the observations shown. These studies will certainly focus more on the differences and therefore on the analysis of shorter ranges of the many years of observations presented here. However, it is anticipated that the causes of the differences will be complex and data sets of limited availability will be needed. Therefore, in our view, such studies differ fundamentally from those presented here and are beyond the scope of this manuscript. We have therefore decided to limit the studies and presentation of the results discussed in this manuscript to the scope presented.

**Responses to the comments of Reviewer 4**

The authors would like to thank the reviewer for his/her positive assessment of the manuscript. In the revised manuscript, all notes listed under "Technical Comments" will be taken into account. We address the reviewer's specific comments below:

1) *It would be useful to include the coordinates of the Andøya radars and of the Davis radar where they are first mentioned in the main body of the text, i.e. on lines 14 and 35, respectively. Although these details are given in the abstract, that is separate from the main text.*

   We have taken up the reviewer's advice and added the corresponding coordinates of Andøya and Davis where they are used for the first time in the text.

2) *The first reference to "mesopause jumps" on line 37 would benefit from slightly more detail. Given that the mesosphere is subject to interhemispheric coupling, it is not immediately clear whether the breakdown of the polar vortex being referred to is in the northern or southern hemisphere. I note that a more detailed explanation is given beginning on line 314.*

   We think that it is not necessary to go into the term "mesopause jumps" in more detail at this point, as a corresponding reference is provided and, as mentioned by the reviewer, the topic will be taken up later in the discussion.

3) *The meaning of abbreviation LMT, which first appears on line 51, should be spelled out. I'm assuming that it refers to Local Mean Time. Moreover, it would be useful to briefly describe what this term means so that it is not confused with the time within the local time zone.*

   In accordance with the reviewer's advice, we have placed the term *Local Mean Time* once in brackets after LMT. An explanation is given in what we consider to be a more appropriate place in the first sentence of paragraph 3.2.2.

4) *The authors use the terms "signal strength" and "volume reflectivity" interchangeably. Although this is somewhat justified, it can be confusing in instances such as on line 132: "1% of the echoes detected by the Davis radar in 2005–2012 period ($Q\_0.01$ ) have a signal strength of $\eta <= 1.6\ 10^{-16}\ m^{-1} \ldots$ ". There are several more similar instances throughout the manuscript.*

   We have made corresponding changes to the text so that in all cases where numerical values of volume reflectivity are listed, these are also linked to this term. In general statements on the signal strength of the echoes, e.g. in the chapter headings, we have left the term signal strength as it is.

5) *I found some of the explanations, at the start of section 3.2.1, of what is being shown in Figure 3 a bit confusing. The following sentence, starting on line 152, was particularly confusing: "The top panels show the mean frequencies of all echo detections normalized to the seasonal maximum value over time and altitude with respect to the daily measurement period." The same point is made much more clearly in the caption for Figure 3, which simply states "The seasonal height distribution of PMSE in the top plots are normalized to its maximum". Conversely, the description from the caption of*

*Figure 3 for what is shown in the middle panels makes it sound as though only a single range gate is being considered. The corresponding text starting on line 155 is clearer. However, I would change the wording slightly to something like "Here, an occurrence is triggered if the volume reflectivity exceeds the minimum threshold at any altitude and at any time during the day".*

We are happy to take the reviewer's advice on notice and have improved caption 3 and the annotated text passages with regard to their comprehensibility.

6) *In the description of first and final PMSE dates in the paragraph beginning on line 161, and in Table 3, it would be of interest if the years during which the earliest/latest occurrences were seen were included as well as the days.*

We have also taken this information on notice and added the exact date of the earliest and latest occurrence in the data records in the table and in the text.

7) *Given that PMSE occurrence is much lower in the first and final months of the season than during the middle two months, in relation to Fig. 5 the authors should state which dates were considered for calculating the mean diurnal variations. For clarification, if PMSEs are deemed to have occurred over 4 km of the 8 km altitude regions considered, does that represent an occurrence rate of 50%? In the context of the middle panels of Fig. 3, this would constitute an occurrence rate of 100%.*

The daily occurrence, as shown in the above figures, refers only to the measurement time and not to a height interval, i.e. 100% would be shown if an echo had been detected in the same range gate in a daily time interval of 6 min on all days on which measurements were taken. In this respect, the seasonal variation, i.e. the lower occurrence of PMSE in the months of May and August, also makes a certain contribution to the diurnal variation. However, the main characteristics, i.e. the maxima and minima, are equally represented in all months, as can be seen, for example, in Fig. 1 in Latteck, et al., 2021. We have included the definition for determining the daily frequency in section 3.2.2.

8) *Also in relation to Fig. 5, it would be useful if the authors could distinguish between single year means (i.e. those represented by the individual black lines) and multi year means (i.e. those represented by the blue and red lines). In some cases in Section 3.2.2, when a reference is made to "mean values", it not immediately clear which type of mean is being referred to.*

We have chosen the term *averaged mean diurnal cycle* or *averaged mean diurnal variation* for the multi-year means and used it in the caption and in the text.

9) *The variable name "$\eta_{min}$" is used in Table 2 to indicate the minimum values in the distributions of volume reflectivities. The same variable name is subsequently used on lines 144 and 248 to indicate the minimum threshold value of volume reflectivity. It would be better to use a name such as "$\eta_{thresh}$" in order to avoid confusion.*

We are happy to accept the reviewer's suggestion and have replaced the term "$\eta_{min}$" with "$\eta_{thr}$" in the appropriate places.

---

## Author Response (AR2)

**Responses to the comments of Reviewer 3**

The authors would like to thank the reviewer for his/her final favourable assessment of the manuscript. All technical corrections were taken into account in the final, revised manuscript:

1) *The terms "peak values" and "peak value of the distribution" (used on line 128) in relation to Fig. 2 confused me. I initially thought they were referring to the maximum value of volume reflectivity rather then to the value of volume reflectivity at which the peak of the distribution occurred. These sentences could be made clearer with something like "with peak values OC-CURRING AT AROUND 6.0 10-15 m-1 and 7.9 10-15 m-1 for Davis and ALWIN, respectively. The CORRESPONDING VALUE for MAARSY is lower at 1.6 10-15 m-1 . . .".*

   The sentence was changed in accordance with the reviewer's suggestion.

2) *A space or a hyphen is needed between the words "twenty" and "four" in the following sentence on line 166: "One twentyfour-minute single-range occurrence is sufficient to trigger the presence of PMSE . . ."*

   The reviewer's proposal was followed.